# Polymer Gels Used in Oil–Gas Drilling and Production Engineering

**DOI:** 10.3390/gels8100637

**Published:** 2022-10-07

**Authors:** Jinliang Han, Jinsheng Sun, Kaihe Lv, Jingbin Yang, Yuhong Li

**Affiliations:** 1School of Petroleum Engineering, China University of Petroleum (East China), Qingdao 266580, China; 2Institute of Engineering and Technology, PetroChina Coalbed Methane Company Limited, Xi’an 710082, China; 3Xi’an Institute of Measurement and Testing Technology, Xi’an 710068, China

**Keywords:** polymer gel, oil–gas drilling and production, crosslinking mechanism, gel properties, application effect, research prospects

## Abstract

Polymer gels are widely used in oil–gas drilling and production engineering for the purposes of conformance control, water shutoff, fracturing, lost circulation control, etc. Here, the progress in research on three kinds of polymer gels, including the in situ crosslinked polymer gel, the pre-crosslinked polymer gel and the physically crosslinked polymer gel, are systematically reviewed in terms of the gel compositions, crosslinking principles and properties. Moreover, the advantages and disadvantages of the three kinds of polymer gels are also comparatively discussed. The types, characteristics and action mechanisms of the polymer gels used in oil-gas drilling and production engineering are systematically analyzed. Depending on the crosslinking mechanism, in situ crosslinked polymer gels can be divided into free-radical-based monomer crosslinked gels, ionic-bond-based metal cross-linked gels and covalent-bond-based organic crosslinked gels. Surface crosslinked polymer gels are divided into two types based on their size and gel particle preparation method, including pre-crosslinked gel particles and polymer gel microspheres. Physically crosslinked polymer gels are mainly divided into hydrogen-bonded gels, hydrophobic association gels and electrostatic interaction gels depending on the application conditions of the oil–gas drilling and production engineering processes. In the field of oil–gas drilling engineering, the polymer gels are mainly used as drilling fluids, plugging agents and lost circulation materials, and polymer gels are an important material that are utilized for profile control, water shutoff, chemical flooding and fracturing. Finally, the research potential of polymer gels in oil–gas drilling and production engineering is proposed. The temperature resistance, salinity resistance, gelation strength and environmental friendliness of polymer gels should be further improved in order to meet the future technical requirements of oil–gas drilling and production.

## 1. Introduction

Polymer gels are polymer materials with a three-dimensional network structure. Their structural pores are usually filled with water, oil or gas. Polymer gels are widely used in flexible electronics, artificial organs, biomimetic materials, biosensors and other fields because of their excellent rheological, mechanical and biocompatibility properties [1]. Since the 1970s, polymer gels have gradually gained popularity and applications in the field of oil–gas drilling and production engineering [2,3]. Oil–gas drilling and production engineering refer to the process of drilling a flow channel between a formation and an oil–gas reservoir and subsequently extracting the buried crude oil and natural gas, bringing it to the surface. Drilling fluids and enhanced oil recovery (EOR) working fluids are two chemical fluids that are required in the oil–gas drilling and production processes [4,5]. It is well known that there are large numbers of pores and fractures in a formation [6]. Drilling fluids can easily leak into the formation through the pores and fractures during drilling, resulting in significant economic losses, blowout, wellbore collapse and other complex accidents [7]. Therefore, one of the most important requirements for safe and efficient drilling and improved oil–gas recovery is the effective plugging of the pores and fractures in formations [8].

Polymer gels have a suitable flexibility and high self-adaptive plugging ability for pores and fractures of different scales in a formation [9]. Therefore, they are widely used in drilling fluids, plugging and lost circulation control in oil–gas drilling processes, as well as in profile control, water shutoff and fracturing [10]. They are one of the most commonly used chemical materials in the field of oil–gas drilling and production engineering. The polymer gel used in oil–gas drilling can be classified as chemically crosslinked gels and physically crosslinked gels based on the different crosslinking methods [11]. Chemically crosslinked polymer gels are gels formed by a covalent bond crosslinking reaction between a crosslinker and a polymer. Physically crosslinked polymer gels are gels formed via hydrophobic associations, hydrogen bonding and other noncovalent crosslinking between polymer molecules [12]. Chemically crosslinked polymer gels have a higher gel strength, better temperature resistance and better salt resistance than other gels used in oil–gas drilling, and they are suitable for high-temperature and high-salt formations [13,14]. Physically crosslinked polymer gels are easy to prepare, environmentally friendly and easy to plug, making them suitable for temporary plugging in the context of oil–gas reservoir protection.

Polyacrylamide (PAM) is the most commonly used polymer for the preparation of gels, whether they are underground or surface crosslinked polymer gels. Polyacrylamide is cheap and has a large number of carboxyl and amide groups [15]. Under high temperatures or alkaline conditions, it can be hydrolyzed to form stable, partially hydrolyzed polyacrylamide (HPAM). The existence of carboxyl and amide groups ensures that it can be cross-linked with various organic/inorganic cross-linkers to form a gel system with good mechanical properties [16]. The most common inorganic crosslinking agent for Cr^3+^ is Al^3+^ high-ionic crosslinking agent. Such metal ion crosslinking agents are cheap, and good results have been achieved in their field application, but the gel system of metal ions, as crosslinking agents, often applies only to reservoirs under 80 °C, and the toxicity of ionic crosslinking agents should not be ignored; thus, their application in high-temperature reservoirs is limited. The commonly used organic cross-linkers include phenolics, polyvinyl imines, etc. [17]. Due to the stronger bond cooperation between covalent bonds, their temperature resistance and mechanical properties are better than those of inorganic cross-linkers, which have a better application range and prospects.

There are two main approaches to exploiting the most commonly used polymer gels, depending on the drilling and production processes: either (a) the prepared gelant is directly injected into the formation and in situ crosslinked into a gel in the formation environment; or (b) after the gelant is pre-crosslinked, the gel particles are dispersed in the liquid phase and injected into the strata [18]. In contrast, in situ crosslinked gels have a high strength, but the gelant is usually affected by the formation shear during injection, causing its strength to be low. Preformed particle gels (PPGs) are not affected by the formation shear, but the accumulation plugging layer formed by the dispersed gel particles in the formation pores cannot match the strength of the overall gel [19].

In this paper, the state of research on in situ crosslinked gels, pre-crosslinked gels and physical crosslinked gels is systematically reviewed from the perspectives of the compositions, cross-linking principles and properties of the gels. In addition to the use of polymer gel materials and the characteristics of polymer gels, the progress in the application of polymer gels in oil and gas drilling and oil and gas production is discussed. Finally, according to the technical requirements of oil and gas drilling and oil production engineering in the future, the research prospects of polymer gel are put forward, which provide a reference for the development of new polymer gel materials that are suitable for oil and gas drilling and oil production.

## 2. Different Types of Crosslinked Polymer Gels

Based on the different application conditions of polymer gels in oil–gas drilling and EOR, the main crosslinking modes of chemically crosslinked polymer gels are in situ crosslinking and surface pre-linking, whereas the main crosslinking modes of physically crosslinked polymer gels are hydrogen bonding, hydrophobic association and electrostatic interaction.

### 2.1. In Situ Crosslinked Polymer Gels

In situ crosslinked polymer gels have been widely used in oil fields because of their remarkable effects in resolving the reservoir heterogeneity, improving the sweep efficiency and increasing the oil washing efficiency. Depending on the crosslinking mechanism, in situ crosslinked polymer gels can be divided into free-radical-based monomer crosslinked gels, ionic-bond-based metal crosslinked gels and covalent-bond-based organic crosslinked gels.

#### 2.1.1. Free-Radical-Based Monomer Crosslinked Gels

Monomer crosslinked gels undergo a rapid gelation process that is initiated by free radicals under the action of crosslinkers. A large number of monomer molecules form polymer gels with a network structure via chain growth and imine crosslinking between chains, improving the bridging effect of crosslinkers [20]. Currently, the commonly used monomers are acrylamide, acrylate, acrylic acid, 2-acrylamide-2-methyl propane sulfonic acid (AMPS), bromoacetic acid (MBAA), tert-butyl acrylate (PAtBA) and other monomers (Table 1).

Acrylamide monomers are one of the most commonly used monomers for gel preparation, and they have been widely studied and applied. Acrylamide and PAtBA are copolymerized by amide bonds formed by the reaction of amine and amide under the action of polyethyleneimine (PEI) (the temperature range of the gel system is 15–175 °C). Zhang et al. [21] systematically investigated the gelation process of acrylamide monomer gels in low-temperature and high-salinity reservoirs, optimized the formulation of the gel system and significantly improved the gelation strength. The gelation time was controlled over 4–48 h, which improved the stability and shear resistance. The gelation process of in situ-formed monomer gels cannot be controlled at high temperatures because it consists of free-radical-initiated rapid processes, and a retardant (potassium ferricyanide) is mostly frequently used to prevent premature gelation in the formation [22]. It is worth noting that acrylamide is assumed to be a carcinogen and a neurotoxin. Therefore, Halliburton Co. developed a less toxic gel system that uses a temperature-activated, acrylate-based monomer [23]. The gelation process in this gel system can be controlled up to a temperature of 93.3 °C, and it provides an improved stability for 1 year at 148.9 °C.

Free-radical-based monomer crosslinked gels can usually be combined with chemical crosslinkers to synthesize in situ monomer crosslinked gels, with acrylic acid as the main crosslinking body, ammonium persulfate as the free radical initiator and chromium (III) salt as the crosslinker (Figure 1) [24]. In addition, nonionic monomers, such as allyl benzoate, diallyl terephthalate, styrene, divinylbenzene, multivinyl monomers and monomer crosslinked gels [25] can be synthesized by free-radical-initiated crosslinking processes.

The gelation strength of monomer gels increases as the monomer concentration increases, implying that a low monomer concentration results in a low gelation strength. Bai et al. [26] discovered that the gelation strength increased as the acrylamide monomer concentration increased during acrylamide monomer synthesis, and the gel structure was the most stable when the monomer concentration reached 15%. The system required a high monomer concentration (4–10%), because the low concentration systems produced gels that were slightly water-soluble and had a relatively low gelation strength [16]. A more cost-effective treatment is required for the oil industry.

#### 2.1.2. Ionic-Bond-Based Metal Crosslinked Gels

Metal crosslinked gels are crosslinked through ionic bonds formed between negatively charged carboxylic acid groups and polyvalent cations. After hydrolysis, metal ions form coordination bonds with hydroxyl groups on the polyacrylamide (PAM) molecular chain, resulting in the formation of a three-dimensional gel network structure. The metal crosslinked gel system has been improved over decades of development. Metal crosslinkers, such as Cr^3+^, Al^3+^, Fe^3+^ and Zr^4+^, can react with carboxyl groups on the polymer chains to form ionic bonds [27].

The most widely used metal crosslinker is chromium metal, which is found in the form of chromium ions. Zhang et al. [28] developed a crosslinker using a combination of chromium acetate and a phenolic resin prepolymer to increase the salt resistance of the polymer/Cr^3+^ gel system to 70,000 mg/L. The combination of a metal ion crosslinker and an organic crosslinker can enhance the performance of gel systems. Cr (III) was crosslinked with ionic bonds formed by the polymers and covalent bonds formed by the phenolic resins and polymers, resulting in a composite crosslinked gel system with a high strength and high-temperature stability (Figure 2) [29]. Chromium acetate/phenolic resin enhanced the gelation strength and maintained the gelation strength at 140 °C for 120 days, without dehydration. The salt tolerance was up to 70,000 mg/L. The temperature resistance of a gel system can be improved by introducing solid-phase inorganic materials into the metal crosslinked gel components and encrypting the three-dimensional gel network structure by filling and embedding it with inorganic materials. Yang et al. [30] used modified silica particles to improve the thermal stability and gelation strength of organic chromium crosslinked gels based on amphoteric polymer synthesis (Figure 3). They demonstrated that a nanosilica surface can adsorb free polymer molecules in a solution and form molecular brushes through charge attraction and hydrogen bonding. The molecular brushes adsorb and combine with the spatial network structure formed by the amphiphilic polymer. Therefore, a gel system can maintain its gelation strength and water content at 85 °C, with a high gelation strength.

Al^3+^ and Zr^4+^ are also commonly used metal gel crosslinkers, in addition to chromium ions. Dai et al. [31] prepared a type of polymer gel using zirconium acetate. The gelation time was controlled over 26–240 h. Reducing the time by 1500s at a shear rate of 2000 rpm could still maintain a 90% gelation strength and good shear resistance. This solves the problem of chromium crosslinked gel systems’ short gelation time and their tendency to be easily affected by the shearing effect of a formation. The crosslinking reaction time of polymer gels synthesized using aluminum citrate/partially hydrolyzed polyacrylamide (HPAM) systems can be controlled by adjusting the aluminum citrate composition. These gel types are suitable for reservoir environments ranging from 20,000 to 50,000 mg/L at 60 °C. Polymer gels based on Al^3+^ crosslinking systems are suitable for medium- and high-temperature reservoirs, while polymer gels based on Cr^3+^ crosslinking systems are suitable for low-temperature reservoirs, owing to differences in reactivity, ligands, crosslinking mechanisms and influencing factors between Cr^3+^ and Al^3+^ [32].

#### 2.1.3. Covalent-Bond-Based Organic Crosslinked Gels

Organic crosslinked gels mainly form three-dimensional network structures through covalent bond crosslinking reactions between the amide group on a polymer and an organic crosslinker [15]. Covalent-bond-based organic crosslinked gels are more adaptive to high-temperature environments than ionic-bond-based metal crosslinked gels and covalent-bond-based metal crosslinked gels because of their higher bond energy and inability to be destroyed [33]. The most commonly used organic crosslinkers include the phenolic and crosslinking polymers (Table 2).

The main crosslinking mechanism of phenolic gels is the condensation reaction between the phenolic and amide groups on polymer molecular chains, which form the body structure. The most common phenolic crosslinkers are phenol/formaldehyde, resorcinol/hexamethylenetetramine (HMTA) and water-soluble phenolic resin [33]. Phenolic crosslinkers can be formed via two approaches. In the first approach, HMTA releases formaldehyde, which then reacts with phenol in the system to form phenolic crosslinkers. In the second approach, phenolic crosslinkers are formed by the polycondensation of A-grade phenolic resins in the system. Phenol undergoes condensation with formaldehyde to produce hydroxymethylphenol or phenolic resin, and then HPAM reacts with hydroxymethylphenol/phenolic resin to produce polymer gels (Figure 4) [34].

Chen et al. [35] synthesized a heat-resistant and salt-resistant polymer gel using phenol, HMTA and resorcinol crosslinkers, as well as acrylamide/AMPS copolymers, and it maintained a gelation strength of 90% after 100 days of aging at a high temperature of 130 °C in a high-salinity environment of 223,000 mg/L, demonstrating excellent stability. The low-toxicity and environmentally protective gel system synthesized using hydroquinone (HQ)/HMTA crosslinkers and PAM can be stably gelated at 120 °C (Figure 5), and the gelation time can be controlled over 5–80 h [36]. This gel system can be used for profile control and water shutoff in high-temperature reservoirs. Organic crosslinked gel systems can be strengthened through the addition of clay. The viscoelasticity and shear resistance of gels prepared by adding bentonite as a modifier increase significantly upon the addition of HMTA organic crosslinkers and HPAM [37].

PEI is commonly used as a crosslinking polymer in oil fields. The amine group in PEI can react with the amide group on the polymer chain to form covalent bonds, resulting in a gel with a three-dimensional network structure. Jayakumar et al. [38] used an HPAM/PEI/AMPS system as a crosslinker and AMPS as a retarding agent to reduce the gelation time of 24–120 h in high-temperature reservoirs at 100 °C. The properties of organic crosslinked gels are affected by high-temperature environments. For example, the viscoelasticity of an acrylamide/PAtBA/PEI gel system is affected by high-temperature environments, and the elastic modulus of the gel system increases by about six times at 100 °C–120 °C [39]. The viscoelasticity of the gel decreases rapidly when the temperature is higher than 150 °C.

### 2.2. Pre-Crosslinked Polymer Gels

Surface chemical pre-crosslinked polymer gel systems were created to solve the problem of the effect of reservoir environments on in situ crosslinked gel systems. By changing the crosslinking mode, gels are formed on the surface and can be injected into the formation as gel particles, eliminating the influence of reservoir environments on gel crosslinking processes. Surface crosslinked polymer gels are divided into two types based on their size and gel particle preparation method, including pre-crosslinked gel particles and polymer gel microspheres.

#### 2.2.1. Preformed Particle Gels

PPGs are formed by polymerizing acrylamide monomers or other monomers with a crosslinker and subsequently drying, crushing and screening the product to obtain gel particles of various sizes [16]. They are injected into the stratum as particles, and they fill and block the pores through bridging accumulation to reduce the permeability. PPGs are affected by the physical and chemical properties of a formation (temperature, pH, etc.) after they enter it. The acidic formation environment has significant influences on the stability and plugging ability of pre-crosslinked gel particles [41]. Zhou et al. [42]. used N, N’-methylene diacrylamide (MBA) and dimethyl diallyl ammonium chloride (DMDAAC) as crosslinkers, as well as acrylamide, based on free radical polymerization, to prepare a gel that can maintain its swelling properties (a swelling rate of 141%), viscoelasticity and shear resistance in an acidic environment with a pH of 3. The influence of the formation environment on the gel particles is inevitable, and high-salinity environments are one of the most common working environments of gel particles. High-salinity formation water destroys the spatial structure of pre-crosslinked gel particles, affecting their strength and swelling properties [43]. Qiu et al. [44] discovered that high-salinity environments (125 °C and 230,000 ppm) have a significant impact on the swelling property of gel particles. The swelling property of the gel particles decreases as the salinity increases, and it becomes stable when the concentration reaches about 10%. Bai et al. [26] synthesized a nanoPPG using acrylamide nanoparticles and discovered that adding nanoclay (Laponite XLG) improved the gel properties, increased the gelation strength by 394% and improved the ability of the gels to withstand adverse and extreme reservoir conditions.

The high temperature of a formation affects the spatial structure of pre-crosslinked gel particles and has negative effects on the strength and stability of the gel particles (Dai et al., 2012). PPG samples absorb heat as the temperature increases, before reaching a sharp exothermic peak. Beyond this point, the PPG structure cannot absorb any additional heat energy that is released later so as to break the chemical bonds (Lenji et al., 2018). Lenji et al. (Abedi Lenji et al., 2018) used an aluminum nitrate nonahydrate crosslinker and PAM to prepare a type of PPG with a good expansion ability and stability in a high-temperature formation at 110 °C. Salunkhe et al. [45] used poly (dimethylacrylamide-sodium co-styrene sulfonate) to develop ultrahigh-temperature-resistant PPGs (HT-PPGs) that can withstand temperatures of 150 °C and pressures of up to 0.7 MPa for at least 18 months without losing their molecular integrity (Figure 6).

#### 2.2.2. Pre-Crosslinked Polymer Gel Microspheres

Polymer microspheres, which are a type of water-expanding particle gel, are generally prepared by reversed-phase suspension polymerization. When injected into the stratum, they can swell up to several times or dozens of times their original volume after entering the high-permeability channel and subsequently block the high-permeability channel. Polymer gel microspheres have a smaller and more uniform particle size than PPGs and have an excellent migration ability in formations [46]. The thermal stability of polymer gel microspheres is one of the essential factors enabling their adaption to high-temperature and complex formation environments. Zhao et al. [47] used acrylamide/AMPS/1-vinyl-2-pyrrolidone (NVP) to prepare terpolymer microspheres with particle sizes ranging from 60 to 90 μm, an excellent temperature resistance and the ability to remain stable at 120 °C. Polymer gel microspheres require an excellent deformation and good shear resistance to achieve deep migration profile control. Low-elastic polymer gel microspheres containing amide, carboxyl and sulfonate groups have been prepared using reversed-phase suspension polymerization, with a swelling rate of 24.5 g/g and shear critical point of 6000 rpm, and they have been widely used for profile control, water shutoff and displacement in heterogeneous reservoirs [48].

New approaches and methods have contributed to the performance optimization of polymer gel microspheres, owing to the development of nanomaterials. The properties of polymer gel microspheres can be improved by employing nanomaterials as reinforcing materials. Silica-reinforced polymer microspheres (Figure 7) were prepared using 3-(methacryl oxide) propyltrimethoxy silane-modified nanosilica as a reinforcing material. The highest degradation temperature was 430 °C, and the average particle size was 20.98 μm. The clogging rate of the polymer gel microspheres reinforced with nanomaterials was increased by 11.54% [49]. Dai et al. [50] synthesized PAM–silica microspheres with core–shell structures using inverse microemulsion polymerization. The maximum degradation temperature was 380 °C, the average pore size was 12 μm and the stable plugging layer was formed at 0.88 MPa.

Pre-crosslinked polymer gel microspheres are widely used in oil–gas drilling and production. They are often used as plugging agents, profile control agents, water blocking agents and displacement control agents in oil fields. Polymer gel microspheres have been developed and improved in terms of their application owing to the demand for instant technology, and they still possess good prospects for development.

### 2.3. Physically Crosslinked Polymer Gels

Physically crosslinked gels are gel network systems with reversible structures and a high performance that are self-assembled through noncovalent bond interactions, as opposed to crosslinked polymer gels, formed by crosslinkers [51]. Physically crosslinked polymer gels are mainly divided into hydrogen-bonded gels, hydrophobic association gels and electrostatic interaction gels depending on the application conditions of oil–gas drilling and production engineering processes [52]. Physically crosslinked polymer gels have several advantages over chemically crosslinked gels, including environmental protection, easy degradation and dynamic reversibility due to the absence of a crosslinker.

#### 2.3.1. Hydrogen-Bonded Physically Crosslinked Gels

Hydrogen bonds are more ubiquitous than other noncovalent bonds. Hydrogen bonds are created with H adjacent to similar groups when N, O and F are contained in the group. However, the instability of hydrogen bonds in water makes the use of hydrogen-bonded physically crosslinked gels challenging [53].

Hydrogen-bonded physically crosslinked gels can maintain stable mechanical properties within certain temperature and pH ranges. However, hydrogen bonding cannot be stable because of interference from water molecules [54]. Traditional single-bond hydrogen bonding can enhance gels through the introduction of hydrophobic groups. Hu et al. [55] used poly (methacrylic acid-co-N/dimethyl acrylamide) (P(MAAc-co-DMAA)) to create a gel (DMAA-co-MAAc) with fracture stress of 2 MPa and a tensile rate of 800%, based on the hydrogen bond formed between the amide group and the acrylic acid group, and then they added the hydrophobic methyl group of PMAAc to stabilize the gel (Figure 8). Gel properties can also be improved with acrylic acid rather than methacrylic acid. A strong and ductile gel (VI-co-AAC) with a fracture stress range of 0.4–1.8 MPa and a ductility of 920–1400% can be obtained by copolymerizing 1-vinylimidazole with acrylic acid to form a strong hydrogen bond between the imidazole and the carboxylic acid group (Figure 9) [56]. Zhao et al. [47] prepared a semi-interpenetrating polymer network (semi-IPN) gel with a layered structure and improved its properties using sodium alginate (SA) and PAM. The hydrogen bond enhanced the self-healing property of the gel (the self-healing efficiency at room temperature was 99%), and the half-layered IPN structure increased the tensile strength of the gel to 266 kPa.

Gels synthesized based on hydrogen bonding usually have a strong ductility and pressure bearing capacity, as well as broad application prospects in the field of oil–gas drilling and production. They can be used as profile control agents and plugging agents to solve water and lost circulation problems in oil–gas drilling and production projects.

#### 2.3.2. Hydrophobic-Association-Based Physically Crosslinked Gels

Hydrophilic monomers are used as the main reaction monomers to form physically crosslinked gels based on hydrophobic associations. They are copolymerized with a small number of monomers with hydrophobic groups, and surfactants are added to form a stable three-dimensional gel network structure [57].

Hydrophobic association gels have different properties depending on the main monomers and hydrophobic monomers used. Acrylamide is used as the main monomer to form the main hydrophilic chain, and two types of octyl phenol polyethoxy ether acrylates (OP-4-AC/OP-10-AC) are used as hydrophobic monomers to form association micelles, which can be used to synthesize a transparent hydrophobic association gel with self-healing abilities [58]. The gelation strength and ductility can be modified by adjusting the hydrophobic monomer content. Hydrophobic association polymer gels usually have a good stability and high toughness because of the presence of hydrophobic groups. Mihajlovic et al. [59] prepared a hydrophobic association gel with a tensile strength of 0.51 MPa, an elongation of 1055% and toughness of 4.12 MJ/m^3^ using hydrophilic polyethylene glycol (PEG) and hydrophobic dimer fatty acid (DFA). The hydrophobic DFA unit self-assembled into a network structure in the micelle domain, which increased the stability of the gel. Xu et al. [60] synthesized a hydrophobic crosslinked gel with favorable properties, such as a 250 kPa tensile strength, 14 MPa compressive strength and 1850% elongation, using acrylamide and an amphiphilic polyblock copolymer, dimethyldodecyl (2-acrylamide-ethyl) ammonium bromide (AMQC1_2_), without a surfactant. Dual physically crosslinked gels based on hydrophobic associations and other physical crosslinking types have a better performance than single hydrophobic-association-based synthetic gels [61]. Yuan et al. [62] synthesized a physically crosslinked gel with a high strength, high toughness and a dual network (DN) using PAM/xanthan gum (XG), which enabled the electrostatic interaction between Ca^2+^ and XG via hydrophobic associations. The fracture stress (3.64 MPa), strain (99%) and compression stress (50 MPa) of the gel were enhanced (Figure 10). Hydrophobic-association-based physically crosslinked gels have a good stability, high toughness, good pressure bearing capacity and good ductility, giving them great application potential in the field of oil–gas drilling and production.

#### 2.3.3. Physically Crosslinked Gels Based on Electrostatic Interaction

Electrostatic interaction is an attractive or repulsive force between groups with positive and negative charges. Non-crosslinked polymer gels synthesized by electrostatic actions can be divided into two types based on their different synthesis mechanisms: (1) gels synthesized by electrostatic interactions between a polyelectrolyte and polyvalent ions with opposite charges, and (2) gels synthesized by electrostatic interactions between two polyelectrolytes with opposite charges [63].

The electrostatic interaction between a fluorine-rich ionic liquid and an ionic liquid was used to synthesize a high-performance ionic gel that can remain stable at 350 °C and has excellent underwater adhesion and self-healing abilities in the pH range of 0–14 [64]. Alginate can also form gels by interacting with ions. Ion exchange reactions with divalent ions, such as Ca^2+^, Sr^2+^ and Ba^2+^, can be used to form SA. Catanzano et al. [65] synthesized a new type of large-pore gel designed for bone tissue engineering using ionic crosslinking between SA and strontium ions. The combination of electrostatic interactions with other noncovalent bonds results in a better-performing ionic crosslinked gel. For example, Fe^3+^ can be introduced to obtain a hydrophobic association and electrostatic dual physically crosslinked gel system (Figure 11) with a fracture stress of 10.39 MPa, which is about 200 times higher than that of a gel system with a single hydrophobic association, based on the hydrophobic association between acrylic acid/acrylamide/octet methacrylate [66]. Tannic-acid-coated cellulose nanocrystals with hydrogen bonds were introduced to obtain a gel system with excellent mechanical properties (strain > 700%) and self-healing ability based on the electrostatic interaction between polyacrylic acid and polyaniline [67]. This gel system has good application prospects in the field of oil–gas drilling and production.

## 3. Polymer Gels for Oil–Gas Drilling

Polymer gel materials are widely used in oil fields to solve various complex problems involved in oilfield development. Different application fields have different action forms (Figure 12). Polymer gels are usually used as plugging agents and drilling fluids in the drilling fluid sector. The drilling of various high-temperature and high-salinity reservoirs has become more difficult because of the demand for reservoir development, and the demand for the use of polymer gels in the drilling fluid sector has also grown.

### 3.1. Polymer Gels as Plugging Agents

Microfractures in formations expand and extend during processes because of the presence of liquid column pressure and capillary forces. Drilling fluids invade the formation through the microfractures and high-permeability reservoirs, resulting in drilling fluid loss and significant formation damage [22]. Adding polymer gel to the drilling fluid as a plugging agent can enhance the ability to plug fractures, prevent pressure transfer, delay fracture propagation and reduce fluid loss [68].

Plugging agents must have a good compatibility, high-temperature resistance, suitable salt resistance and strong plugging ability, as well as the qualities of easy degradation and environmental protection. Preformed crosslinked gels are used as plugging agents based on the principle of particle blocking bridging, and they are used selectively depending on the particle size and fracture compatibility. Zhu et al. [69] developed a degradable PPG (DPPG) with excellent hydrophilicity (Figure 13). This gel can rapidly swell by absorbing water after entering the fracture channel and subsequently fill and plug the fracture channel to achieve the goal of plugging, and the plugging pressure can reach 21.12 MPa. In regard to the microscale and nanoscale pores of shale formations and high-temperature environments, Li et al. [70] synthesized a nanoscale plugging agent using a gel material with a double crosslinking structure, which can be adsorbed on shale surfaces through hydrogen bonding and electrostatic interactions and can quickly and effectively fill shale pores and establish a plugging layer at high temperatures of 150–200 °C. Xu et al. [71] developed nanosized polymer microspheres that are suitable for water-based drilling fluids and can enter and fill the pores and fractures of shales under pressure, form a plugging layer and still retain more than half of the particle size after aging at high temperatures (150–200 °C) in order to keep the plugging layer stable (Figure 14).

In recent years, nanomaterials have been widely used and developed in the oilfield chemistry sector, providing a new direction for the improvement of polymer gels as plugging agents. A nanogel plugging agent was prepared by introducing magnesium silicate nanoparticles into the HPAM/chromium acetate (Ш) system. It can enhance the adhesion force of the gel and fracture walls and thus strengthen the intensity of the blocking layer [72].

### 3.2. Polymer Gels as Lost Circulation Materials

Lost circulation is a common problem in drilling engineering. Poor treatment delays the progress of a project and cause economic losses [73]. The effect of conventional plugging materials is not ideal because of the complexity of in situ strata, and it is easy to generate false plugging in the case of large fractures [74]. Polymer gels are one of the chemical lost circulation materials commonly used in oil–gas drilling engineering. They can form high-strength plugging layers, block fractures and stabilize well walls. They have had a positive effect on field applications in recent years [75].

When dealing with different lost circulation conditions, it is necessary to select the appropriate control method and the appropriate lost circulation material. Surface chemical pre-crosslinked gels are often used as lost circulation materials for minor lost circulation (Figure 15) [42]. Kang et al. [76] developed preformed sawdust gel particles. Under the effect of pressure difference, the particles preferentially entered high-permeability channels to plug the fractures or high-permeability channels. In addition, the particles had a suitable viscoelasticity and deformation ability. They could increase the depth of lost circulation control through deformation and migration. Nanomaterials have been used to improve the performance of crosslinked gels. The new silica-nanoparticle-based strengthened dispersed particle gels can be used in three-phase foaming systems to improve the foam migration stability in fractures and pores and can also be aggregated and bridged in fractures to form a stable plugging layer [77].

In situ chemically crosslinked gels are widely used to address the problem of lost circulation in fractured formations (Figure 16). However, the performance of gel lost circulation materials is easily affected by high temperatures, high pressures and high salinity. Liu et al. [78] developed a high-temperature (120–200 °C) resistant gel with a relatively small three-dimensional network structure (less than 5 μm). The gel system was able to maintain most of its initial viscosity and viscoelasticity even after being subjected to mechanical shear or porous medium shear. The gel system exhibited an excellent adhesion ability in the formation pore medium and could adapt well to the pore structure, indicating that it has a strong anti-scouring ability. This gel system can block most of the pores in porous media. Li et al. [79] created a high-temperature, high-strength (HTHS) gel system that can gel at high temperatures (150 °C), has a good expansibility and can fill fractures. Moreover, the gel has a good expansion property under high-temperature conditions, allowing it to bond with fractures more effectively and produce a high-strength plugging layer with a blocking pressure of up to 10 MPa.

Physically crosslinked gels can also be used as lost circulation materials to tackle loss in fractured formations. The physically crosslinked gel (ZND) was developed, in which the polymer chain spontaneously accumulates through intermolecular hydrophobic associations in aqueous solutions, forming a reversible dynamic physically crosslinked network structure, filling fractures and forming a “gel slug” that can isolate the fluid in the formation from that in the wellbore, thereby achieving the goal of lost circulation control [80].

## 4. Polymer Gels for Oil–Gas Production

Most domestic oil fields use water injection technology for reservoir exploitation. However, reservoirs face complex problems, such as a high water cut and low oil recovery, as they progress through the middle and late stages of development. Polymer gels have been widely used in the field of oil–gas production because of their exceptional performance. They are mainly used as profile control agents, water shutoff agents and flooding agents.

### 4.1. Polymer Gels as Profile Control Agents

Due to reservoir heterogeneity, China’s primary method of oil field development is water drive development, which requires a large amount of water to be injected into oil fields during exploitation [81]. The use of profile control technology can effectively control the oil–water ratio, modify the water absorption profile of the water injection layer and improve the oil recovery [82]. A polymer gel, which is a common type of profile control agent used in oil fields, can be adsorbed in the pore throat and change the water absorption profile.

In situ crosslinked polymer gels have a low viscosity before gelling and can deeply penetrate formations and certain tiny fractures [83]. Owing to the permeable polymer gel network, adding a retarder to the traditional gel system can prolong the gelation time to 6–7 h under low-temperature conditions (30 °C), increase the migration time of the gel in the formation and have a positive application effect on the oil saturation reduction near the wellbore area of low-temperature fractured reservoirs [84]. The mixed gel can easily be filtered during formation injection and migration, reducing the quality of the polymer gel. The anti-shearing property of the first crosslinked gel was enhanced by its hyperbranched structure, rigid core and hydrophobic pendants. The weak crosslinked structure of the first crosslinked gel effectively minimized block solution filtrate loss and matrix damage, allowing the first crosslinked gel to enter the fracture and enabling the formation of the secondary crosslinked gel in order to control the profile [85].

For profile control agents, crosslinked gel particles must have a good shear and expansion resistance, the ability to be retained or adsorbed after entering the permeable layer and the ability to be intercepted and bridged in the pore throat or large-pore channel [54]. Zhao et al. [86] prepared a phenolic resin dispersed granular gel with a good temperature resistance and shear resistance. This gel can be retained, adsorbed, bridged and blocked after entering the formation, thereby reducing water production in the high-permeability layer and improving the formation profile. It has been successfully applied in the Changqing oil field, with an effective period of more than 500 days. The diameter size of preformed crosslinked gel particles affects their formation in terms of the migration effect. Submicron- to micron-grade phenolic crosslinked polymer gels (Figure 17) can gather in a large-pore space or directly block the pore throat to plug a high-permeable zone (Figure 18). Dispersed particle gels (DPG) can achieve in-depth profile control because of the elastic deformation and migration of reservoir porous media [40]. The temperature resistance of the gel can be improved, and the plugging layer can be stabilized at 200 °C by employing phenolic resin, formed by combining resorcinol and formaldehyde as a crosslinker, and by using oxalic acid and ammonium chloride as stabilizers, thereby enhancing the plugging ability of the profile control agent. The plugging rate of sandstone in filling simulations is above 98% (Han et al., 2020a).

### 4.2. Polymer Gels as Water Shutoff Agents

Some oil wells are prone to water problems due to formation heterogeneity, different fluid flows, long-term water flooding and casing damage, all of which have severe effects on the economic benefits and production life of oil wells [87]. Profile control and water shutoff are common methods for enhancing the oil recovery in water flooding oil fields. Water shutoff agents have been widely used in oil fields [88]. Polymer gels have been widely used as water shutoff agents because of their simple preparation, low cost and excellent performance.

Conventional gel plugging agents have the limitations of a slow gelation, weak strength and instability in low-temperature and high-salinity reservoirs. Zhang et al. [21] divided the crosslinking reaction process of acrylamide monomer gels into three phases: the induction, rapid crosslinking and stable phases. The induction phase was extended by changing the contents of the monomer and crosslinker to ensure the gel flow and migration in the fractures. The gel can produce effective plugging in high-permeability channels, with a plugging rate of more than 96%. The reaction between resorcinol and HMTA produces phenolic resin, which has a closeknit network structure. By changing the HMTA concentration, the gel crosslinking time can be adjusted, and the gel strength in the fractures can be improved [89]. The unique hydrophobic association and low rate of pyrolysis of amphiphilic polymer gels reduce the influence of high-temperature and high-salinity environments (80 °C and 80,000 mg/L) on the plugging layer established by gel slug in fractures, as well as the loss of polymer molecule adhesives on the rock surfaces caused by gel fluid flowing through the sand particles [90].

The migration flowing behavior of the gelant in a fracture can be divided into three streams: fracture flow, leak-off flow and matrix flow. These behaviors cause the gel to become distributed in three different patterns after gelling: gel clusters in the fracture, gel layers on the fracture surface and dispersed gel lumps in the matrix pores and throats (Figure 19) [21]. As the reservoir heterogeneity increases, gel water shutoff agents with a high initial viscosity lose their ability to enter medium- and low-permeability layers in order to form gel clusters and thus cannot meet the requirements for deep-water shutoff. Huang et al. [91] proposed a low-initial-viscosity gel system (the initial viscosity is less than 10 mPa·s) with a long gelation time and minor pollution of the middle- and low-permeability layers. It can thoroughly penetrate fractures and form gel clusters, thereby meeting the requirements for deep plugging. Polymer gels have been widely used as water shutoff agents in oil fields. The performance requirements of polymer gel are gradually increasing as the complexity of water shutoff problems increases.

### 4.3. Polymer Gel as Chemical Flooding Agents

In recent years, chemical flooding has become an important technical method for improving the oil recovery in oil fields affected by water flooding in the middle and late stages [92]. The difference between the injection pressure and formation absorption pressure is increased by increasing the injection pressure of medium- and low-permeability layers and large pore throats, thereby increasing the sweep coefficient [93]. Polymer gel flooding agents, which are widely used in oil fields, have a good high-pressure-bearing capacity, suitable plugging performance and suitable migration ability in formations [94].

Traditional polymer gel materials cannot penetrate deep permeable layers because of the limitation of their particle size, causing the pollution of low-permeability layers. Weak gel systems have a low strength, high adsorption capacity and high capture capacity in porous media, and they are easily retained in porous media, preferentially entering high-permeability layers and reducing pollution in low-permeability layers [95]. In order to improve the oil recovery, Cui et al. [96] optimized an ultrahigh-molecular-weight HPAM/water-soluble phenolic weak gel system, which reduced the reservoir heterogeneity and the shear effect of the small pore structures on the gelant and enhanced the plugging effect of the weak gel, in the case of the large pores, by aggregating the oil droplets in the small pores to form large oil droplets when they passed through the narrow throat and moved out of it in the form of oil flow. Polymer microspheres are also one of the most commonly used chemical flooding agents. Nanoscale polymer gel microspheres (30–60 nm) can adsorb, aggregate and bridge in pore channels, reducing the water permeability [97]. They can also deeply penetrate the oil reservoirs to achieve a high flooding efficiency.

Nanogels have also been developed in the field of modulated flooding owing to recent advances in nanotechnology. Nanogels can be attached to contact surfaces between fluids in order to reduce interfacial tension [98]. Nanogels can be used to improve the oil recovery by reducing interfacial tension and changing the wettability of rocks. Roussennac et al. [99] investigated Brightwater^®^, a submicron polymer gel with a delayed swelling rate that is stimulated to expand by multiple times under high temperatures in deep formations, thereby blocking the permeable layer (Figure 20).

### 4.4. Polymer Gels as Fracturing Fluids

Hydraulic fracturing is a common technical method for improving the oil recovery in oil fields. The key aspect of this technology is the selection of a suitable fracturing fluid [100]. Polymer gels are a common type of hydraulic fracturing fluid used in oil fields [101]. The addition of polymer gels to fracturing fluids increases their viscosity, improves their ability to transport proppant and reduces fluid loss.

Guar gum (GG) is a gel that has been widely used in oil field fracturing fluid systems to increase the viscosity of fracturing fluids and minimize fracturing fluid loss (Figure 21). Hydroxypropyl guar gum (HPG) and carboxymethyl guar gum (CMG) have different microstructures and viscoelastic properties. HPG gels have a higher fiber density, better suspension ability and slower settling velocity than CMG gels [102]. The novel fluorinated hydrophobically associating cationic guar gum gel (FCGG), which is based on GG and cationic guar gum (CGG), has a better heat resistance (85 °C) and shear resistance than conventional fracturing gels, and it also has good application prospects in the fracturing fluid industry [103].

Physically crosslinked gel systems based on intermolecular noncovalent bond interactions also have many applications in the fracturing fluid sector. It is possible to form hydrogen bonds between the surface of titanium dioxide and the hydroxyl group of HPG. The crosslinking reaction between TiO_2_ nanoparticles produced by the hydrolysis of the Ti complex and HPG can increase the viscosity of HPG gelants by 25 times [104]. Jiang et al. [105] combined supramolecular gel fracturing fluids with fiber materials to form a mechanical network structure in the fluids and an interweaved supramolecular honeycomb network structure in order to enhance the fluid load-bearing and consolidation capacity, thus improving the proppant suspension. This gel has many application prospects in oil fields. Polymer gel fracturing fluids are also evolving and improving in tandem with the evolution of fracturing fluid requirements. Polymer gels have broad application prospects as fracturing fluids because of continuous technological advancements.

## 5. Prospects of Polymer Gels for Oil–Gas Drilling and Production

Polymer gels are commonly used materials in the field of oil–gas drilling and production engineering, drilling fluids and EOR. In this context, polymer gels tend to maintain or even improve the working efficiency of the operation in increasingly complex and difficult environments. Currently, the main development and improvement directions for polymer gels are their temperature resistance, compressive properties, rheological properties, gelation strength, environmental protection performance and environmental responsiveness.

### 5.1. Improving the Temperature Resistance

The temperature resistance of polymer gels is the most basic and important index. Reservoir temperatures are increasing, and polymer gel temperature resistance requirements are increasing as deep and ultra-deep oil–gas reservoirs are developed [106]. Currently, the main solutions to this problem can be divided into three directions: the selection of a crosslinker, the selection of a crosslinking polymer or monomer and the addition of new chemical agents. The type of crosslinker used has an important influence on the temperature resistance of a polymer gel. The use of metal–organic mixed crosslinkers can improve the temperature resistance of polymer gel systems. For example, chromium lactate and water-soluble phenolic resin were used as crosslinkers to improve the thermal stability of a gel system. There was no noticeable dehydration after 30 days of aging at 200 °C (Figure 22) [107]. In general, organic crosslinkers are better than metal crosslinkers at high temperatures, because covalent bonds are more stable at high temperatures than ionic bonds [108]. Adding organic monomers to a polymer structure can inhibit the thermal hydrolysis of acrylamide. For example, the combination of phenol/formaldehyde and a proper polymer can improve the temperature resistance of a gel system [109]. The temperature resistance of a gel system can also be improved by adding resorcinol and formaldehyde to the polymer gel system and generating phenolic resins [110].

High-temperature-resistant chemicals can be used to improve the overall temperature resistance of polymer gels during the crosslinking process. For example, a new type of composite polymer gel with a good thermal stability can be prepared by adding SiO_2_ and inducing surface activity in the original polymer gel system [111]. The abovementioned methods are all based on the improvement of the gelation mechanism of a crosslinked gel. Through the development of physically crosslinked gels, experts and scholars have discovered that noncovalent-bond-based physically crosslinked gels have a good temperature resistance and excellent developmental prospects. Polymer gels have a good temperature resistance, as a new type of temperature-resistant material. Thus, any increase in disposable energy will inevitably increase the cost. Therefore, a major issue for the development of polymer gels is determining how to improve their thermal resistance while reducing the development costs.

### 5.2. Improving the Salinity Resistance

Reservoir environments are often highly saline, and their formation water contains a variety of inorganic salts. These environments can have an adverse effect on the performance of polymer gels, reducing the efficiency of the operation. Although the salt resistance of polymer gels has been studied and improved over the years, inorganic salts still affect polymer gels.

The influence of a high-salinity environment on the performance of a polymer gel is mainly evident in the gel structure. Under the influence of cations, the electric potential of the polymer surface decreases, the molecular structure curls and shrinks and the swelling performance of the gel is greatly reduced [112]. The crosslinker chosen for in situ crosslinked gels must be capable of stable crosslinked gelling in high-salt-content environments and minimizing the influence of the environment on the gelation strength. Resorcinol, HMTA and oxalic acid can be used as modifiers to improve the salt resistance of polymer gels (stable gelling at 280,000 ppm) [89]. Calcium alginate gels composed of acrylamide, MBAA, APS, SA and nanoclay have the quality of high-salt resistance (soaking in artificial seawater for 30 days keeps the systems stable) [112]. Surface crosslinked polymer gels prevent the crosslinking environment from affecting the crosslinking process and, instead, use a preformed crosslinking process. The effect of the salt resistance of the gel is mainly manifested by the process of the gel entering the reservoir environment as a working fluid. Silica nanoparticles have been used to fill voids in three-dimensional gel networks so as to maintain the stability in a high-salt reservoir environment (212,000 ppm) [77]. Non-crosslinked polymer gels do not require a crosslinker in the crosslinking process. They are formed by hydrogen bonds, intermolecular forces and hydrophobic associations, and they are less affected by high-salinity environments and have a good salt resistance. The stability of the gel systems can be maintained in a pH range of 0–14 via ion–ion interactions between ionic liquids [64].

Improving the salt resistance of polymer gels, particularly physically crosslinked gels, has always been an urgent demand of the field. The goal of research and development has always been to obtain the best salt resistance at the lowest cost. The use of raw materials with an improved salt resistance can also be beneficial, but for crosslinked gels, an improved crosslinker can provide more demonstrable and richer benefits.

### 5.3. Improving the Rheological Property

Rheological properties play an important role in the injection and migration of polymer gels in a formation. When a polymer gel is used as a drilling fluid, its rheological properties affect its ability to carry cuttings and reduce filtration. When a polymer gel is used as a plugging agent, its rheological properties will affect its drainage capacity and filling capacity as a plugging agent. The rheological properties will affect the viscosity and viscoelasticity of the solvent. The rheology determines whether in situ crosslinked polymer gels can migrate to the appropriate position. Weak subsurface crosslinked gel systems have a low initial viscosity (<30 mPa·s) and good compressive deformation and strength, and can migrate to deep strata [113]. Controlling the initial viscosity of surface crosslinked polymer gels can ensure a smooth gel injection process. The rheological properties of a gel also affect its migration when it enters a formation. Nanomaterials can change the low-end rheology properties of gel drilling fluids while maintaining a low plastic viscosity [114]. Adding nanosilicon materials into a gel system can enhance the strength of the prefabricated particle gel, improve the viscoelasticity of the gel system and improve the rheological properties of the gel system (under 1000% strain, the elastic modulus is still significant, without fracture) [115]. Both the modification of the original materials and the development of new materials require time to accumulate in order to improve the rheological properties of the gels to varying extents.

### 5.4. Improving the Gelation Strength

Gelation strength is an important factor that affects the formation of a high-strength plugging layer by a polymer gel [43]. Polymer gels with a high plugging strength are often used as drilling fluids, plugging agents, profile control agents and water shutoff agents. Many factors affect their gelation strength. The molecular weight of the polymer largely determines the gelation strength, and the concentration of the crosslinker plays a key role in determining the gelation strength after gelling [116]. In general, the gelation strength increases as the polymer molecular weight and crosslinker concentration increase [117]. The crosslinking environment affects the gelation strength of an in situ gel, and the high temperature and high salinity of the reservoir destroys the formation of the three-dimensional network structure of the crosslinked gel, thereby reducing the gelation strength. HPAM, methyl hydroxybenzoate and triethylenetetramine can be used to improve the high-temperature resistance (150 °C) and high-pressure-bearing capacity (the plugging strength reaches 0.25 MPa/cm in a 5 mm fracture) of a gel.

The gelation strength of a gel increases when its temperature resistance and salt resistance are improved. The crosslinking process of surface gels is controllable and not affected by reservoir environments. In addition to the influence of the crosslinker on the gelation strength, the influence of the crosslinking process is significantly reduced. The gelation strength has been substantially increased (by 394%) with the development of nanomaterials [118]. The strength and degradability of a gel can be improved by introducing nanosilica into the conventional crosslinking system (the bearing pressure can reach 13–20.5 MPa for a core with a fracture width of 0.398–0.170 cm) [119]. Physically crosslinked gels based on noncovalent bonds have a good gel-forming strength. A high-strength and high-toughness physically crosslinked gel with a DN was synthesized by combining electrostatic action and hydrophobic association. The fracture stress (3.64 MPa), strain (99%) and compression stress (50 MPa) of the gel were enhanced (Yuan et al., 2016). The key to the application of a polymer gel, whether it is a chemically crosslinked gel or a physically crosslinked gel, is to improve its gelation strength as much as possible while keeping its cost low.

### 5.5. Improving the Environmental Friendliness

The environmental friendliness of a polymer gel mainly refers to the protection of the reservoir environment. The retention of a polymer gel in a formation mainly reflects its pollution of the formation. The gel degrades after entering the formation, but it is still retained in the formation. Gel materials are toxic, and most crosslinkers are toxic chemical products. Therefore, a polymer gel also becomes toxic after gelation, which has a negative impact on the formation. The crosslinking process of in situ crosslinked polymer gels is carried out in the formation, and the crosslinker and polymer residue pollute the formation. Many metal crosslinkers are toxic, and this type of polymer gel can affect a reservoir environment when it remains in the formation. The environmentally friendly polymer gels based on zirconium acetate have the quality of high-salt resistance and shear resistance (the gelation strength retention rate remains above 90% at 1500 rpm after shearing at 2000s), making it suitable for water shutoff treatment in low-temperature reservoirs [31].

During migration, surface crosslinked gel particles pollute the low-permeability formation layer, thereby reducing the permeability, preventing oil displacement agents from entering and reducing the recovery. Degradable prefabricated gel particles prepared using acrylamide and AMPS as raw materials leave less residue and have less influence on a formation after degradation. Polymer gel microspheres have been developed to effectively reduce the formation damage of a low-permeability layer during gel particle migration. Polymer gel microspheres have small particle sizes and can enter deep strata, and their pollution of the low-permeability layer is less significant than that of conventional gel materials. Low-elastic polymer gel microspheres have a suitable deformation resistance, a suitable shear resistance and a minor effect on the formation [120]. Physically crosslinked gels do not depend on the presence of crosslinkers during the gelling process, and their gelling mechanism is based on intermolecular forces, hydrogen bonds and hydrophobic associations, all of which have minor influences on the formation. However, they have not been applied on a large scale because of technical problems. The difficulty in studying the environmental performance of polymer gels is determining how to achieve maximum environmental protection without negatively affecting the other properties of the gels and polluting the formation. Additionally, cost control is a major factor, as increasing costs to improve the environmental performance of gels has certain benefits and drawbacks.

## 6. Conclusions

Polymer gels have been improved significantly since their creation. They can be divided into crosslinked polymer gels and non-crosslinked polymer gels depending on the crosslinking form. Crosslinked polymer gels can be divided into in situ crosslinked polymer gels and crosslinked polymer gels depending on the different crosslinking sites. In situ crosslinked polymer gels can be divided into the monomer crosslinked type, metal crosslinked type, organic crosslinked type and preformed crosslinked type. Preformed crosslinked types can be subdivided into pre-crosslinked gel particles and polymer gel microspheres. Non-crosslinked polymer gels can be divided into the hydrogen bonding type, hydrophobic association type and electrostatic force type based on the different gelling methods. Polymer gels have been widely used in the field of oil–gas drilling and production because of their excellent performance. Polymer gels are commonly used as plugging agents and drilling fluids in the drilling fluid sector. Polymer gels can be classified as profile control agents, water shutoff agents, flooding agents and fracturing fluids according to their different mechanisms in the EOR sector.

Polymer gels should be developed in order to improve their ability to adapt to complex reservoir environments and the effects of their application in these environments. Since the different types of polymer gels have different application effects, future development directions should be distinguished. Preformed crosslinked polymer gels are generally used in the drilling fluid sector. When used as a plugging agent, they must have a good compatibility, high-temperature resistance, suitable salt resistance and strong plugging ability, with the qualities of easy degradation and environmental protection. Plugging agents should have a good swelling performance, stability, viscoelasticity and shear resistance. Drilling fluid should have good temperature and salt resistances, rheological properties and shear resistance. In situ crosslinked polymer gels have numerous applications in oil–gas production. Preformed crosslinked polymer gels have begun to acquire more applications as a result of their advancements. They must have a good expansion performance, long crosslinking time, strong migration ability in the formation, high gelation strength and high shear resistance when used as profile control agents. Water shutoff agents must have a good temperature resistance and salt resistance. Polymer gels must have good rheological properties when used as flooding agents, as well as a good suspension ability when used as fracturing fluids.

In view of the harsh conditions of high temperatures and high salt contents in an increasing number of oil and gas reservoirs, in order to expand the application scope of polymeric gels, the properties of high-temperature resistance, high-salt resistance and rheological/mechanical properties of gels are mainly improved by improving the structures and types of polymers, crosslinkers and additives. Here, the mechanisms of polymer gels used for reservoir profile control and water plugging, deep formation control and flooding, drilling plugging and other applications were analyzed, providing theoretical and technical references for the development and optimization of polymer gels that are suitable for different oil and gas drilling and production processes.

## Figures and Tables

**Figure 1 gels-08-00637-f001:**
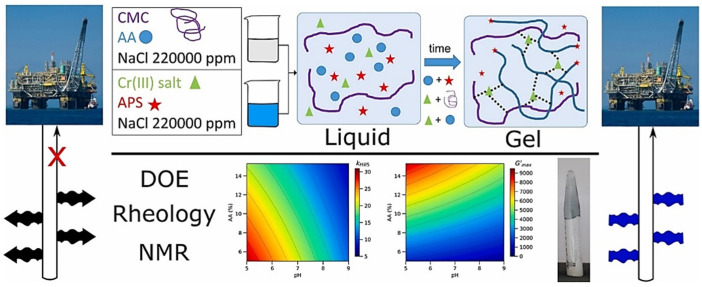
Mechanism diagram of free-radical crosslinking combined with acrylic acid-based chemical crosslinking. (Reprinted/adapted with permission from Ref. [24], 2022, Pereira et al.).

**Figure 2 gels-08-00637-f002:**
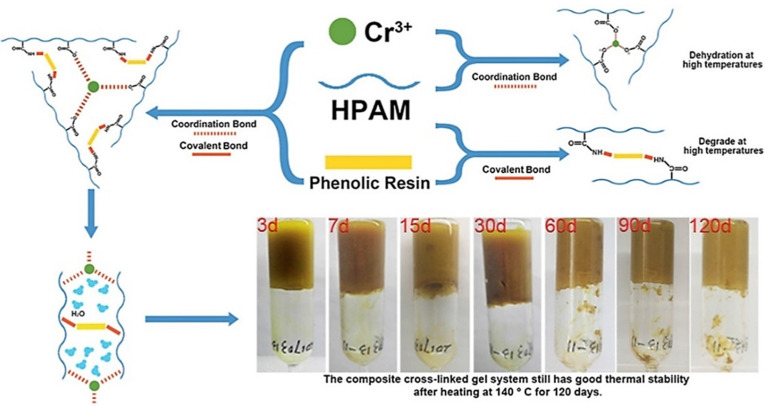
Crosslinking mechanism of Cr (III) and phenolic resin. (Reprinted/adapted with permission from Ref. [29], 2020, Zhang et al.).

**Figure 3 gels-08-00637-f003:**
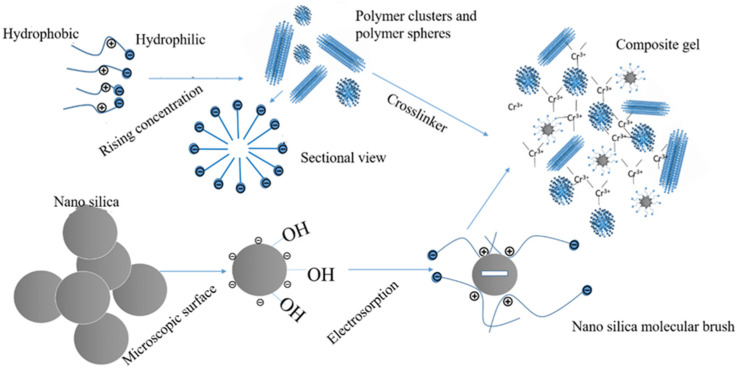
Mechanism diagram of organic chromium gel reinforced by nanosilica. (Reprinted/adapted with permission from Ref. [30], 2019, Yang et al.).

**Figure 4 gels-08-00637-f004:**
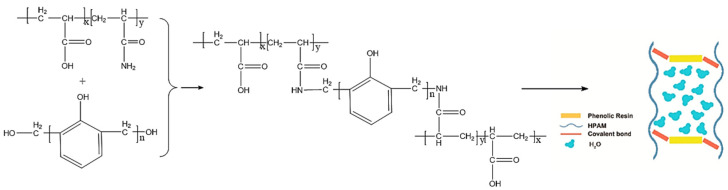
Crosslinking mechanism of phenolic resin. (Reprinted/adapted with permission from Ref. [34], 2020, Zhang et al.).

**Figure 5 gels-08-00637-f005:**
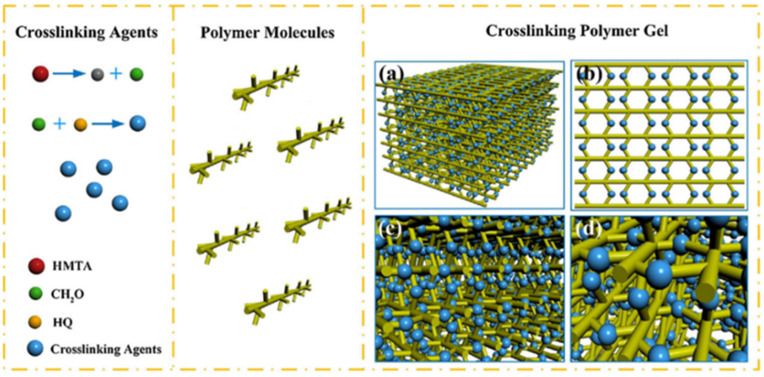
Crosslinking mechanism of HQ/HMTA. (**a**–**d**) Three-dimensional network structure of gels from different perspectives. (Reprinted/adapted with permission from Ref. [40], 2016, Liu et al.).

**Figure 6 gels-08-00637-f006:**
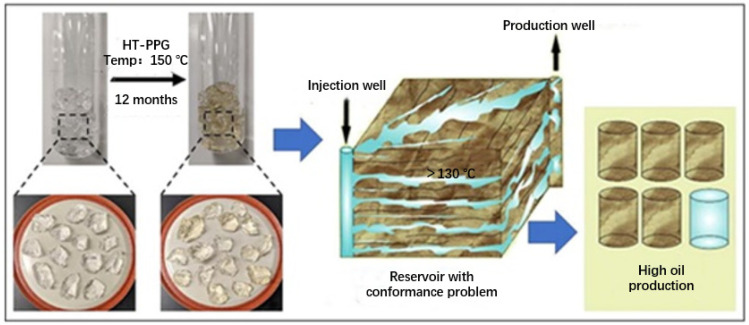
Simulation of the high-temperature aging test and formation application of HT-PPGs. (Reprinted/adapted with permission from Ref. [45], 2021, Salunkhe et al.).

**Figure 7 gels-08-00637-f007:**
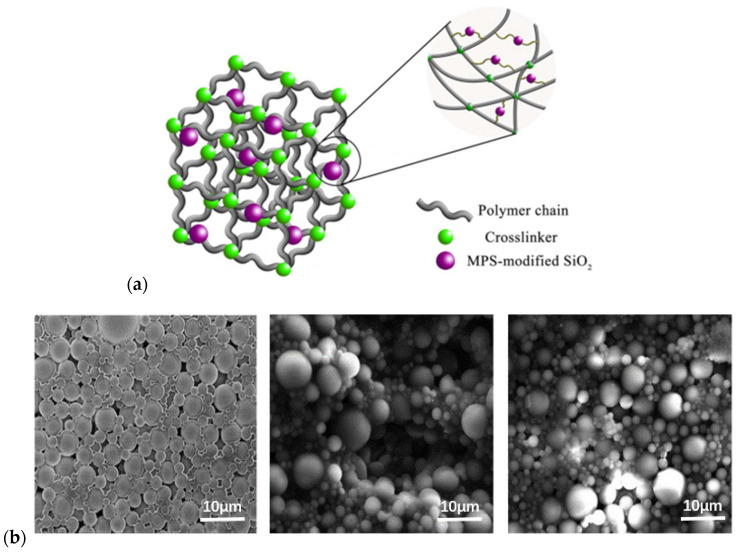
Structure diagram of polymer gel microspheres: (**a**) structure of silica-reinforced polymer microspheres and (**b**) microstructure of polymer microspheres. (Reprinted/adapted with permission from Ref. [49], 2018, Tang et al.).

**Figure 8 gels-08-00637-f008:**
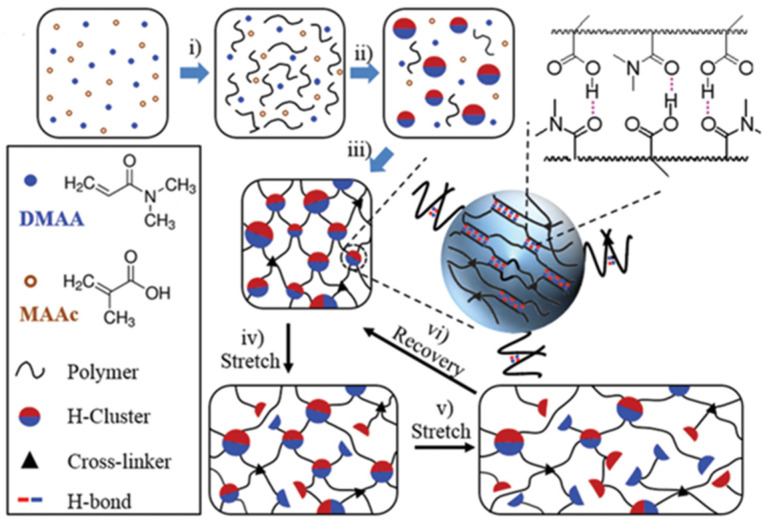
Preparation mechanism of DMAA-co-MAAc gels. (Reprinted/adapted with permission from Ref. [55], 2015, Hu et al.).

**Figure 9 gels-08-00637-f009:**
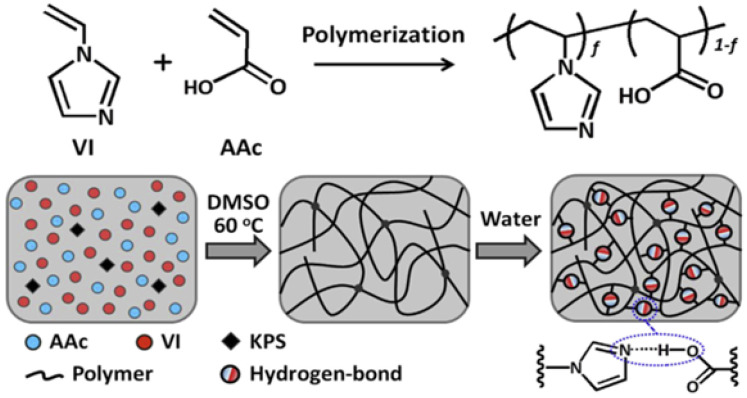
Preparation mechanism of 1-vinylimidazole-co-acrylic acid (VI-co-AAc) gels. (Reprinted/adapted with permission from Ref. [56], 2017, Ding et al.).

**Figure 10 gels-08-00637-f010:**
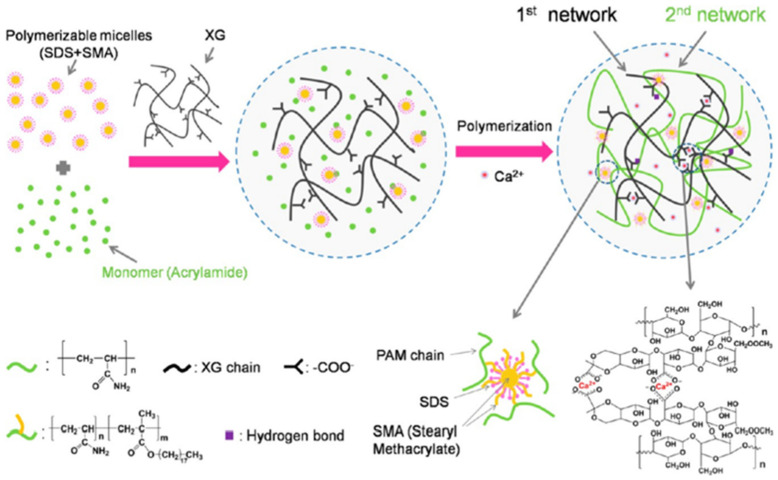
Preparation mechanism and network structure of PAM/XG hydrogels. (Reprinted/adapted with permission from Ref. [62], 2016, Yuan et al.).

**Figure 11 gels-08-00637-f011:**
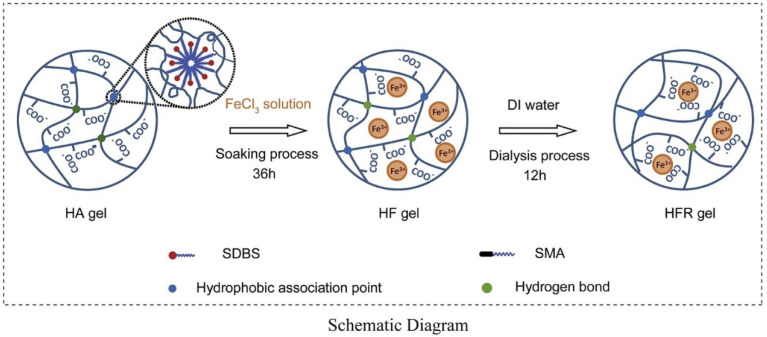
Synthesis mechanism of dual physically crosslinked gels based on hydrophobic associations and electrostatic interactions between iron ions. (Reprinted/adapted with permission from Ref. [66], 2018, Hu et al.).

**Figure 12 gels-08-00637-f012:**
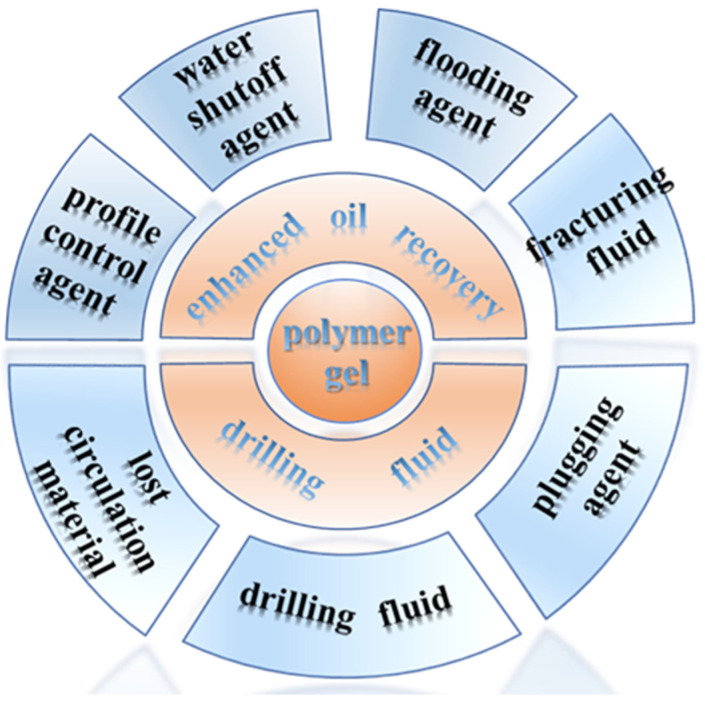
Application of polymer gels in oil–gas drilling engineering.

**Figure 13 gels-08-00637-f013:**
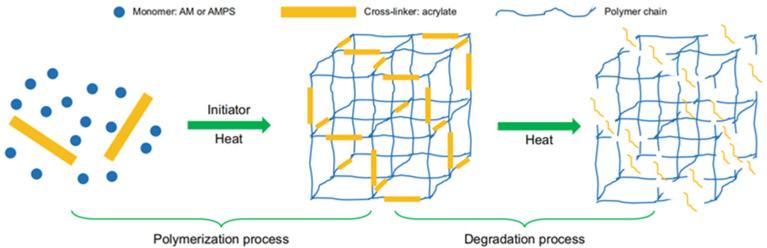
Degradation mechanism of DPPGs. (Reprinted/adapted with permission from Ref. [69]. 2021, Zhu et al.).

**Figure 14 gels-08-00637-f014:**
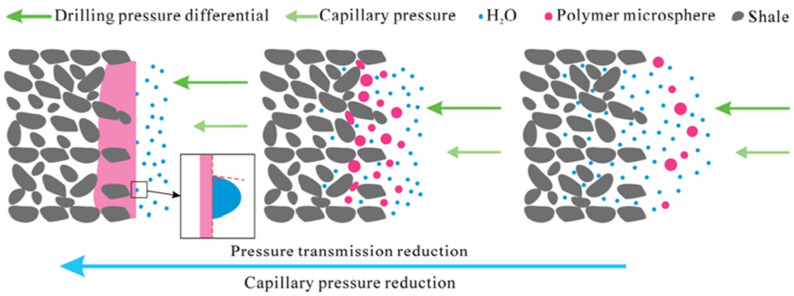
Pore and fracture plugging mechanism of nanoscale polymer microspheres for shales. (Reprinted/adapted with permission from Ref. [71], 2018, Xu et al.).

**Figure 15 gels-08-00637-f015:**
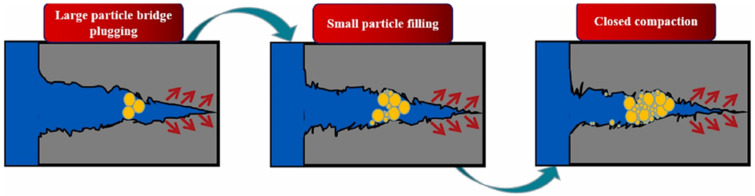
Plugging mechanism of PPGs in fractures. (Reprinted/adapted with permission from Ref. [42], 2022, Zhou et al.).

**Figure 16 gels-08-00637-f016:**
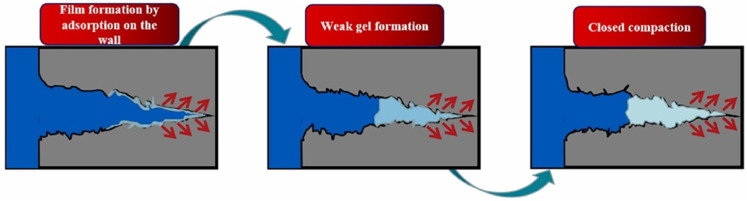
Plugging mechanism of in situ crosslinked gel slug in fractures. (Reprinted/adapted with permission from Ref. [42], 2022, Zhou et al.).

**Figure 17 gels-08-00637-f017:**
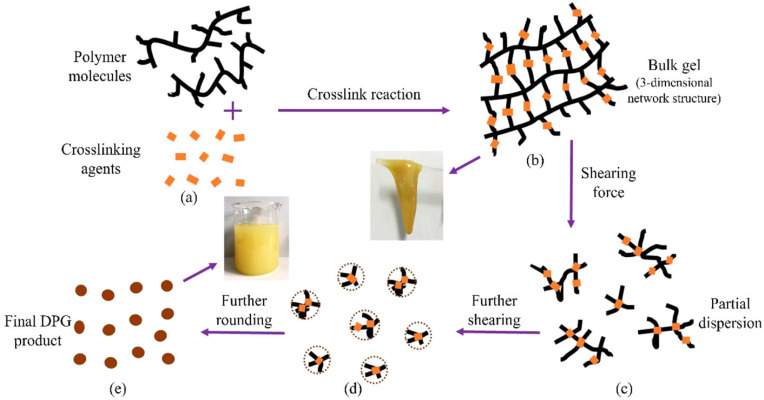
Preparation mechanism of DPGs. (**a**–**e**) the cross-linking reaction process of the bulk gel and the shearing process using the colloid mill. (Reprinted/adapted with permission from Ref. [40], 2016, Liu et al.).

**Figure 18 gels-08-00637-f018:**
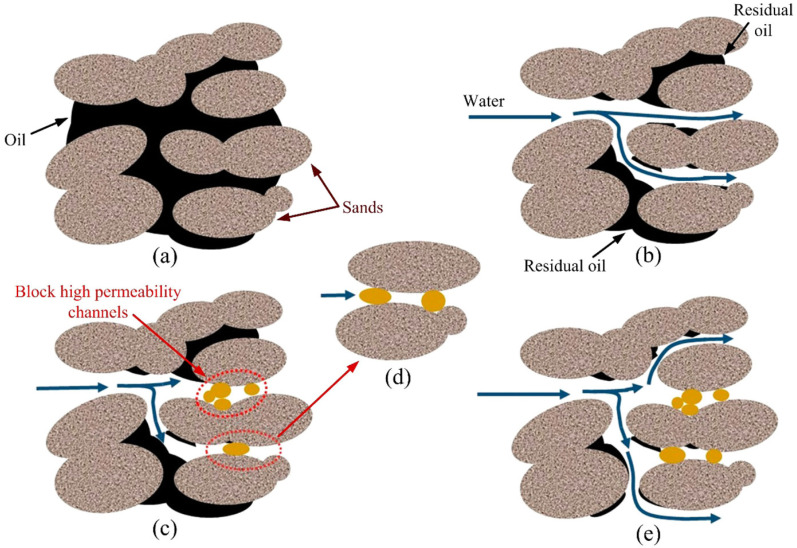
Profile control mechanism and microstructure of DPG. (**a**) Initial distribution of crude oil in the reservoir; (**b**) water breakthrough along the high permeability channels after long-term water flooding; (**c**) plugging the high permeability channels by DPG; (**d**) deformed DPG particles pass through the pore throat; (**e**) increased sweep efficiency after the DPG treatment. (Reprinted/adapted with permission from Ref. [40], 2016, Liu et al.).

**Figure 19 gels-08-00637-f019:**
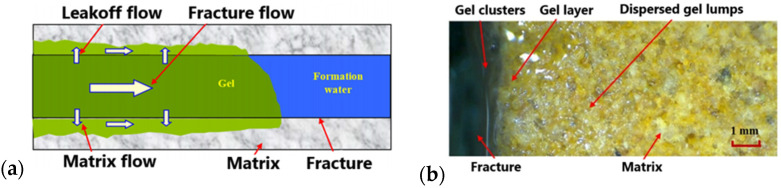
Simulation and example of flow behaviors of gelants in fractures: (**a**) migration and flow of gelants in fractures and matrices; (**b**) overall distribution pattern of polymer gels in fractures and matrices after gelation. (Reprinted/adapted with permission from Ref. [21], 2020, Zhang et al.).

**Figure 20 gels-08-00637-f020:**
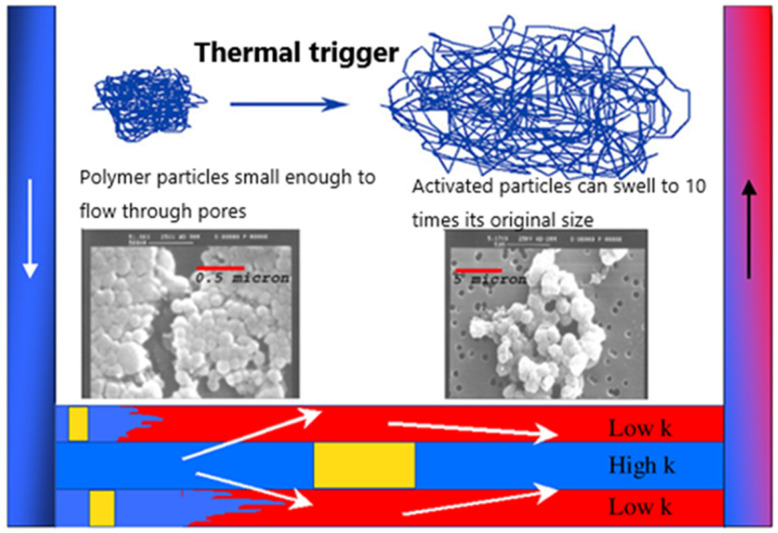
Flooding mechanism of Brightwater^®^. (Reprinted/adapted with permission from Ref. [99], 2010, Roussennac and Toschi.).

**Figure 21 gels-08-00637-f021:**
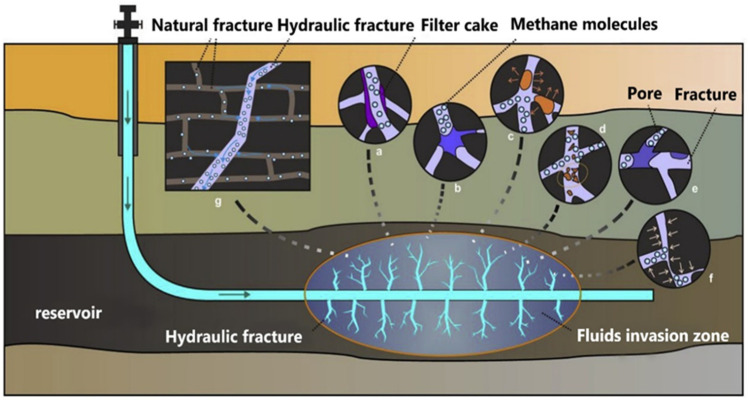
Fracturing stimulation and fracturing fluid damage: (**a**) compacted filter cakes are formed on the fracture surface, generated by fracturing. (**b**) After the filtrate enters the pore throat, the particles block the channel. (**c**) The filtrate causes the clay to expand, disperse and migrate, blocking the pores. (**d**) Some residual solid particles may cause throat blockage. (**e**) When the filtrate enters the fracture, it forms a concentrated liquid, preventing the gel from breaking. (**f**,**g**) GG adsorbs onto the fracture surface, resulting in a decrease in the porosity and permeability. (Reprinted/adapted with permission from Ref. [91], 2019, Huang et al.).

**Figure 22 gels-08-00637-f022:**
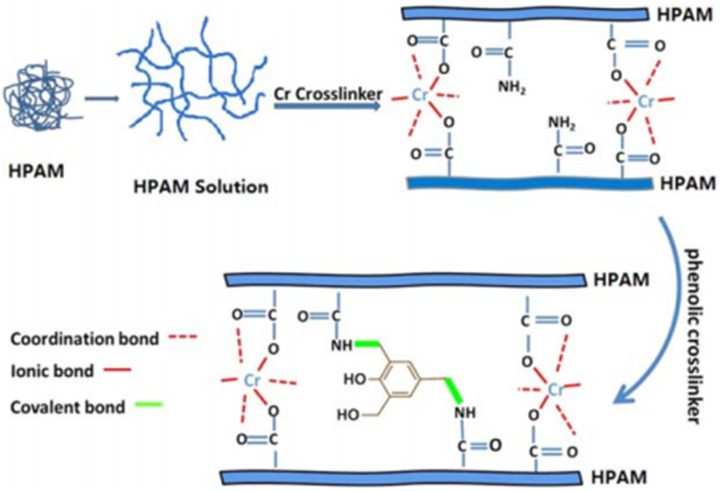
Crosslinking mechanism between chromium lactate and water-soluble phenolic resins. (Reprinted/adapted with permission from Ref. [107], 2015, Zhang et al.).

**Table 1 gels-08-00637-t001:** Commonly used monomers.

No.	Name	Molecular Formula	Structural Formula
1#	Acrylamide	C_3_H_5_NO	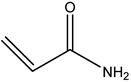
2#	Acrylate	CH_2_ = CHCOOR	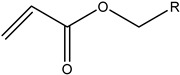
3#	Acrylic acid	C_3_H_4_O_2_	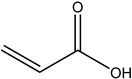
4#	2-Acrylamido-2-methylpropane sulfonic acid (AMPS)	C_7_H_13_NO_4_S	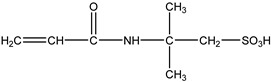
5#	Bromoacetic acid (MBAA)	C_2_H_3_O_2_Br	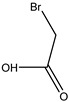
6#	Tert-butyl acrylate (PAtBA)	C_7_H_12_O_2_	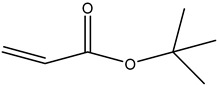

**Table 2 gels-08-00637-t002:** Commonly used crosslinkers for covalent organic crosslinked gels.

No.	Name	Molecular Formula	Structural Formula
1#	Phenol	C_6_H_5_OH	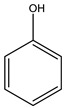
2#	Formaldehyde	HCHO	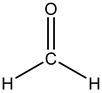
3#	Resorcinol	C_6_H_6_O_2_	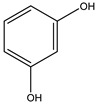
4#	Hexamethylenetetramine (HMTA)	C_6_H_12_N_4_	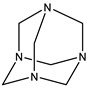
5#	Water-soluble phenolic resin	C_7_H_6_O_2_	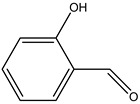
6#	Polyvinyl imine	(CH_2_CH_2_NH) n	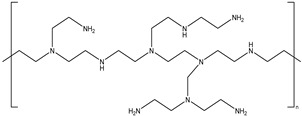

## Data Availability

All persons included have agreed to confirm.

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
