# Peer review of "Polymer Gels Used in Oil–Gas Drilling and Production Engineering"

_gels, 2022, doi:10.3390/gels8100637_

Round 1

Reviewer 1 Report

The article is comprehensive and good, but it needs corrections to be published in the journal.

1. The abstract of the present study needs a basic rewriting and a quantitative analysis report.

2. This group has presented a review study in 2022 under this title. What is the necessity of conducting the present study??

Types and Performances of Polymer Gels for Oil-Gas Drilling and Production: A Review. Gels8(6), 386.

3. The innovation of the study should be described in detail. What will this study help researchers and industrialists in the future? State the industrial applications of this study?

4. The introductory literature is in dire need of revision and recent and applied studies should be included in it.

5.  What does figure 1 indicate? Is it not good quality?

6. The quality of Figure 4 is not good. It seems to be cut from other people's articles?

7. Please provide a copy of the originality and dissimilarity of this study?

8. The quality of figure 5 is not good at all

9. The scale is not specified in Figure 8.

10. Figure 11 is not clear. It seems to be cut from other people's studies.

11. Improve the quality and clarity of Figure 20.

12.  All symbols used in the table and abbreviations should be reported in tabular format.

Author Response

Reviewer 1

The article is comprehensive and good, but it needs corrections to be published in the journal.

Comment 1. The abstract of the present study needs a basic rewriting and a quantitative analysis report.

Response: Thanks for your careful review of our manuscript. We have revised and supplemented this paragraph in detail and rewritten it. Detailed revisions were marked the red in the revised manuscript.

Comment 2. This group has presented a review study in 2022 under this title. What is the necessity of conducting the present study??

Types and Performances of Polymer Gels for Oil-Gas Drilling and Production: A Review. Gels, 8(6), 386.‏

Response: Thanks for your careful review of our manuscript. This paper mainly reviews the application of polymer gels in drilling engineering. Previous publications have focused on the types and performances of polymer gels. Therefore, the emphasis of the two articles is different.

Comment 3. The innovation of the study should be described in detail. What will this study help researchers and industrialists in the future? State the industrial applications of this study?

Response: We feel great thanks for your professional review work on our manuscript. Your comments are critical to improve the novelty of our articles. We have made detailed revisions and supplements in the revised manuscript. Detailed revisions were marked the red in the revised manuscript.

Comment 4. The introductory literature is in dire need of revision and recent and applied studies should be included in it.

Response: We feel great thanks for your professional review work on our manuscript. Your careful review and comments have greatly improved our article. Based on your suggestions, we have extensively expanded each section, supplementing it with corresponding case studies and descriptions of research findings from the literature. Detailed revisions and supplements were marked the red in the revised manuscript.

Comment 5. What does figure 1 indicate? Is it not good quality?

Response: Thanks for your comment. According to your suggestion, we decided to remove Figure 1 after comprehensive consideration.

Comment 6. The quality of Figure 4 is not good. It seems to be cut from other people's articles?

Response: We feel great thanks for your professional review work on our manuscript. Your careful review and comments have greatly improved our article. We have redrawn the picture according to the reference article. Detailed revisions and supplements were marked the red in the revised manuscript.

Comment 7. Please provide a copy of the originality and dissimilarity of this study?

Response: Thanks for your careful review of our manuscript. Your comments are critical to improve the novelty of our articles. We have made detailed revisions and supplements in the revised manuscript. Detailed revisions were marked the red in the revised manuscript.

Comment 8. The quality of figure 5 is not good at all

Response: Thanks for your comment. We have redrawn the picture according to the reference article. Detailed revisions and supplements were marked the red in the revised manuscript.

Comment 9. The scale is not specified in Figure 8.

Response: We feel great thanks for your professional review work on our manuscript. We have marked the corresponding scale in the corresponding position of the article picture.

Comment 10. Figure 11 is not clear. It seems to be cut from other people's studies.

Response: We feel great thanks for your professional review work on our manuscript. Your careful review and comments have greatly improved our article. We have redrawn the picture according to the reference article. Detailed revisions and supplements were marked the red in the revised manuscript.

Comment 11. Improve the quality and clarity of Figure 20.

Response: Thanks for your careful review of our manuscript. We have redrawn the picture according to the reference article. Detailed revisions and supplements were marked the red in the revised manuscript.

Comment 12. All symbols used in the table and abbreviations should be reported in tabular format.

Response: Thanks for your careful review of our manuscript. Your earnest attitude and meticulous spirit are worthy for our study. We have revised the article in detail according to your professional comments. All the abbreviations introduced in the whole manuscript are described and explained in detail. Detailed revisions were marked the red in the revised manuscript.

Reviewer 2 Report

The review is well written. Some issues need to be addressed by authors prior to acceptance for publication:

1. Some confusing phrases are detected in abstract. Need to revise. In addition, conclusion can be added.

2. It is good to include relevant figure, photo or diagram to support the introduction.

3. “Drilling fluids can easily leak into the formation through the pores and fractures during drilling, resulting in significant economic losses, blowout, wellbore collapse, and other complex accidents (Ahmad et al., 2021). However, EOR working fluids easily flow along the pores and fractures during the production process, reducing oil–gas recovery (Abou-al-fitooh et al., 2021).” This part in first paragraph is confusing. EOR is supposedly to increase oil and gas production, not to reduce.

4. On polymer gel in drilling application, the review is shallow and ambiguous. In fact, the examples given for drilling application are not convincing. It is not common to use polymer gel in drilling operation. The polymer (not polymer gel) as mud cake enhancer and to reduce loss of circulation is more widely used.

5. Plugging agent is not introduced during drilling operation. It will damage the formation.

6. Basically, profile control and water shut off are under same umbrella, which is under conformance control. Profile control is also to reduce or control water production same as water shut off.

7. The manuscript is written in the way application of polymer and polymer gel have been mixed up and misleading. Polymer gel is not applicable for all application in oil and gas. Polymer gel has a specific role especially in profile modification. It is suggested to revise the manuscript accordingly.

8. The conclusion also needs to be rewritten. The current form is too brief and not specifically address the objective of review.

9. There are some typos, grammatical error, improper sentence structures, inaccurate terminology used in this review that need to be proofread and corrected.

10. The length of paragraphs which should not be too short or too long.

Author Response

Reviewer 2

The review is well written. Some issues need to be addressed by authors prior to acceptance for publication:

Comment 1. Some confusing phrases are detected in abstract. Need to revise. In addition, conclusion can be added.

Response: Thanks for your careful review of our manuscript. We have revised and supplemented this paragraph in detail and rewritten it. Detailed revisions were marked the red in the revised manuscript.

Comment 2. It is good to include relevant figure, photo or diagram to support the introduction.

Response: We appreciate your constructive comments and suggestions, which have been helpful during the revision and improvement of our work. We have added this part of content in the introduction of the revised manuscript.

Comment 3. “Drilling fluids can easily leak into the formation through the pores and fractures during drilling, resulting in significant economic losses, blowout, wellbore collapse, and other complex accidents (Ahmad et al., 2021). However, EOR working fluids easily flow along the pores and fractures during the production process, reducing oil–gas recovery (Abou-al-fitooh et al., 2021).” This part in first paragraph is confusing. EOR is supposedly to increase oil and gas production, not to reduce.

Response: We feel great thanks for your professional review work on our manuscript. We are sorry for the misunderstanding due to our incorrect expression and accept your advice. Detailed revisions and supplements were marked the red in the revised manuscript.

Comment 4. On polymer gel in drilling application, the review is shallow and ambiguous. In fact, the examples given for drilling application are not convincing. It is not common to use polymer gel in drilling operation. The polymer (not polymer gel) as mud cake enhancer and to reduce loss of circulation is more widely used.

Response: Thanks for your careful review of our manuscript. We accept your valuable advice. Your comments are of great help to improve the quality of the manuscript. We reviewed the application of polymer gels in drilling with accurate examples. Detailed revisions and supplements were marked the red in the revised manuscript.

Comment 5. Plugging agent is not introduced during drilling operation. It will damage the formation.

Response: Thanks for your comment. We accept your valuable advice. Your comments are of great help to improve the quality of the manuscript. Detailed revisions and supplements were marked the red in the revised manuscript.

Comment 6. Basically, profile control and water shut off are under same umbrella, which is under conformance control. Profile control is also to reduce or control water production same as water shut off.

Response: We appreciate your constructive comments and suggestions, which have been helpful during the revision and improvement of our work. We have revised and added this part of content in the revised manuscript.

Comment 7. The manuscript is written in the way application of polymer and polymer gel have been mixed up and misleading. Polymer gel is not applicable for all application in oil and gas. Polymer gel has a specific role especially in profile modification. It is suggested to revise the manuscript accordingly.

Response: Thanks for your careful review of our manuscript. We apologize for the misunderstanding caused by our carelessness. In view of your comments, we re-organized the logical structure, the content writing and the improvement in order to make the structure of the article more rigorous. Detailed revisions and supplements were marked the red in the revised manuscript.

Comment 8. The conclusion also needs to be rewritten. The current form is too brief and not specifically address the objective of review.

Response: Thanks for your comment. Your comments are critical to improve the quality of our articles. We have supplemented the information in the revised manuscript. Detailed revisions and supplements were marked the red in the revised manuscript.

Comment 9. There are some typos, grammatical error, improper sentence structures, inaccurate terminology used in this review that need to be proofread and corrected.

Response: Thanks for your careful review of our manuscript. We are sorry for the misunderstanding due to our unclear expression. We carefully revised the terminology in the article by referring to relevant resources. And read the full text, the grammar of the whole article has been revised and improved in detail in the revised manuscript. Detailed revisions were marked the red in the revised manuscript.

Comment 10. The length of paragraphs which should not be too short or too long.

Response: Thanks for your careful review of our manuscript. We have revised and supplemented this paragraph in detail and rewritten it. Detailed revisions were marked the red in the revised manuscript.

Reviewer 3 Report

The review manuscript is written comprehensibly and in a well arranged way. The authors describe the application of different types of polymer gels for oil–gas drilling and production engineering. Three gel systems are discussed; the in-situ covalently crosslinked gels, the physical gels and the surface crosslinked gels in the form of particles or microspheres. The synthesis of the gels and their properties with respect to the  oil–gas drilling and other factors related to the topic are described. The most important factors of the field are thoroughly discussed and the required properties of gels to be efficient are explained. The various types of gels are compared and their advantages or disadvantages are highlighted.

The manuscript covers a broad range of the topic including the latest literature results. The reader will receive a good survey of the corresponding field.

I have only minor objections.  

The authors should unify the physical units used in the manuscript. The temperature in Fahrenheit (line 108) or the pressure in psi units (line 267).

The formula of 2-acrylamido-2- methylpropane sulfonic acid (AMPS) in Table 1 is incompletely given.

The hydrogen atom is missing in the acrylic acid (AAc) formula in Fig.10.

Also the misprint in the title (Engneering) should be corrected.

I recommend to publish the manuscript after a minor revision.

Author Response

Reviewer 3

The review manuscript is written comprehensibly and in a well arranged way. The authors describe the application of different types of polymer gels for oil–gas drilling and production engineering. Three gel systems are discussed; the in-situ covalently crosslinked gels, the physical gels and the surface crosslinked gels in the form of particles or microspheres. The synthesis of the gels and their properties with respect to the oil–gas drilling and other factors related to the topic are described. The most important factors of the field are thoroughly discussed and the required properties of gels to be efficient are explained. The various types of gels are compared and their advantages or disadvantages are highlighted. The manuscript covers a broad range of the topic including the latest literature results. The reader will receive a good survey of the corresponding field. I have only minor objections. I recommend to publish the manuscript after a minor revision.

Comment 1. The authors should unify the physical units used in the manuscript. The temperature in Fahrenheit (line 108) or the pressure in psi units (line 267).

Response: Thanks for your careful review of our manuscript. Your earnest attitude and meticulous spirit are worthy for our study. We have revised the article in detail according to your professional comments.

Comment 2. The formula of 2-acrylamido-2- methylpropane sulfonic acid (AMPS) in Table 1 is incompletely given.

Response: Thanks for your careful review of our manuscript. Your earnest attitude and meticulous spirit are worthy for our study. We have made detailed revisions and supplements in the revised manuscript. Detailed revisions were marked the red in the revised manuscript.

Comment 3. The hydrogen atom is missing in the acrylic acid (AAc) formula in Fig.10.

Response: We feel great thanks for your professional review work on our manuscript. Your careful review and comments have greatly improved our article. We have made detailed revisions and supplements in the revised manuscript.

Comment 4. Also the misprint in the title (Engneering) should be corrected.

Response: Thanks for your careful review of our manuscript. We have made detailed revisions and supplements in the revised manuscript. Detailed revisions were marked the red in the revised manuscript.

Reviewer 4 Report

The authors developed a comprehensive review paper on the application of polymers in different upstream operations. They complemented their work with critical discussion and future recommendation. This paper could serve as a good reference for future researchers interested in this area. 

- I recommend running a final check on the English writing for minor mistakes such as "filed" in line 19. It should have been "field".

- I recommend checking for any copyright issues for the cited figures.

- I recommend adding some paragraph on the use of molecular simulation to engineer polymers, and to assess the performance of polymers under subsurface conditions. Examples of such work are given below:

1- 10.3390/gels8070442

2- https://doi.org/10.1115/1.4054785

Author Response

Reviewer 4

The authors developed a comprehensive review paper on the application of polymers in different upstream operations. They complemented their work with critical discussion and future recommendation. This paper could serve as a good reference for future researchers interested in this area.

Comment 1. I recommend running a final check on the English writing for minor mistakes such as "filed" in line 19. It should have been "field".

Response: Thanks for your careful review of our manuscript. We are sorry for the misunderstanding due to our unclear expression. We carefully revised the terminology in the article by referring to relevant resources. And read the full text, the grammar of the whole article has been revised and improved in detail in the revised manuscript. Detailed revisions were marked the red in the revised manuscript.

Comment 2. I recommend checking for any copyright issues for the cited figures.

Response: Thanks for your comment. In this article, cite the source of the data in the form of either "Data from..." or "Adapted from...". We will supplement it as required. Detailed revisions were marked the red in the revised manuscript.

Comment 3. I recommend adding some paragraph on the use of molecular simulation to engineer polymers, and to assess the performance of polymers under subsurface conditions. Examples of such work are given below:

1- 10.3390/gels8070442

2- https://doi.org/10.1115/1.4054785

Response: Thanks for your careful review of our manuscript. We have revised and supplemented this paragraph in detail and rewritten it. Detailed revisions were marked the red in the revised manuscript.

Round 2

Reviewer 2 Report

As a substantial improvement has been made, this manuscript is acceptable for publication. Just need further improvement on language, grammar and referencing format.